# Closed-loop neuromodulation restores network connectivity and motor control after spinal cord injury

Patrick D Ganzer[1,2]*, Michael J Darrow[1], Eric C Meyers[1,2], Bleyda R Solorzano[2], Andrea D Ruiz[2], Nicole M Robertson[2], Katherine S Adcock[3], Justin T James[2], Han S Jeong[2], April M Becker[4], Mark P Goldberg[4], David T Pruitt[1,2], Seth A Hays[1,2,3], Michael P Kilgard[1,2,3], Robert L Rennaker II[1,2,3]*

[1]Erik Jonsson School of Engineering and Computer Science, The University of Texas at Dallas, Richardson, United States; [2]Texas Biomedical Device Center, Richardson, United States; [3]School of Behavioral Brain Sciences, The University of Texas at Dallas, Richardson, United States; [4]Department of Neurology and Neurotherapeutics, University of Texas Southwestern Medical Center, Dallas, United States

*For correspondence:
patrick.ganzer.neuro@gmail.com
(PDG);
renn@utdallas.edu (RLRII)

**Abstract** Recovery from serious neurological injury requires substantial rewiring of neural circuits. Precisely-timed electrical stimulation could be used to restore corrective feedback mechanisms and promote adaptive plasticity after neurological insult, such as spinal cord injury (SCI) or stroke. This study provides the first evidence that closed-loop vagus nerve stimulation (CLV) based on the synaptic eligibility trace leads to dramatic recovery from the most common forms of SCI. The addition of CLV to rehabilitation promoted substantially more recovery of forelimb function compared to rehabilitation alone following chronic unilateral or bilateral cervical SCI in a rat model. Triggering stimulation on the most successful movements is critical to maximize recovery. CLV enhances recovery by strengthening synaptic connectivity from remaining motor networks to the grasping muscles in the forelimb. The benefits of CLV persist long after the end of stimulation because connectivity in critical neural circuits has been restored.
DOI: https://doi.org/10.7554/eLife.32058.001

## Introduction

Recovery from serious neurological injury requires substantial rewiring of neural circuits. Many methods have been developed to enhance synaptic plasticity in hopes of enhancing recovery. Unfortunately, these methods have largely failed in the clinic likely due to the challenge of precisely targeting specific synapses and absence of testing in clinically-relevant models (*Gladstone et al., 2006*; *Levy et al., 2016*). Real-time control of neural activity provides a new avenue to promote synaptic plasticity in specific networks and restore function after injury (*Engineer et al., 2011*; *Nishimura et al., 2013*; *McPherson et al., 2015*; *Guggenmos et al., 2013*).

Human and animal studies demonstrate that precisely timed vagus nerve stimulation (VNS) can improve recovery of sensory and motor function. VNS engages neuromodulatory networks and triggers release of pro-plasticity factors including norepinephrine, acetylcholine, serotonin, brain-derived neurotrophic factor, and fibroblast growth factor (*Hays, 2016*; *Hulsey et al., 2017*; *Hulsey et al., 2016*). This in turn influences expression and phosphorylation of proteins associated with structural and synaptic plasticity, including *Arc*, CaMKII, TrkB, and glutamate receptors (*Alvarez-Dieppa et al., 2016*; *Furmaga et al., 2012*). Engagement of neuromodulatory networks activates a transient synaptic eligibility trace to support spike-timing-dependent plasticity (STDP) (*He et al., 2015*), thus raising

**eLife digest** The spine houses a network of neurons that relays electric signals from the brain cells to the muscles. When the spine is injured, some of these neurons may be damaged and their connections to the muscles broken. As a result, the muscles they command become weak, and movement is impaired. It is possible to strengthen the remaining neural connections with physical rehabilitation, but the results are limited.

Vagus nerve stimulation, VNS for short, is a new technique that could help people recuperate better after their spine is injured. The vagus nerve controls the heart, lungs and guts, and it reports the state of the body to the brain. When this nerve is electrically stimulated, it releases chemicals that can strengthen the neural connections between brain, spine and muscles, and even create new ones. This rewiring process is essential to repair or bypass the broken neural connections caused by a spine injury. However, it is still not clear how best to use VNS to optimize recovery.

Here, Ganzer et al. study how VNS helps rats whose forelimbs are weakened after a spine injury. Three groups of rats go through physical rehabilitation, using their affected front paws to pull a handle and feed themselves. Two of these groups also receive VNS: their vagus nerve is stimulated either after the best trials (strongest pulls) or worst trials (weakest pulls).

Compared to the rehab-only and the worst trials-VNS animals, the rats that receive VNS on the best trials while using their affected paw have many more neuronal connections between their brain and the muscles in this limb. These muscles also become much stronger. VNS during the movement improves recovery whether the rodents have one or two front limbs injured, and the benefits are long lasting.

As the rats pull the handle, the neurons involved in the movement get activated: they then carry a molecular 'signature' that lasts for a short time. When VNS is applied during that window, it appears to help these neurons form new connections with each other, as well as strengthen existing ones. These improved connections mean the brain can communicate better with the muscles: movement is enhanced, which results in greater functional recovery compared to rehabilitation alone.

VNS is already trialed in stroke patients, who have weakened muscles because their brain neurons are damaged. The work by Ganzer et al. provides crucial information on how VNS could ultimately improve the recovery and quality of life of people with spine injuries.
DOI: https://doi.org/10.7554/eLife.32058.002

the prospect that closed-loop neuromodulatory strategies may provide a means to direct specific, long-lasting plasticity to enhance recovery after neurological injury. Indeed, in the absence of neurological damage, repeatedly pairing sensory or motor events with brief bursts of VNS yields robust plasticity in sensory or motor cortex that is specific to the paired experience (*Engineer et al., 2011*; *Hulsey et al., 2016*). Moreover, the addition of VNS to rehabilitative training improves recovery in rodent models of unilateral brain injury and in chronic stroke patients, highlighting the clinical potential of closed-loop neuromodulatory strategies (*Hays, 2016*; *Khodaparast et al., 2016*; *Pruitt et al., 2016*; *Hays et al., 2014a*; *Dawson et al., 2016*).

We tested the hypothesis that closed-loop VNS (CLV) could be harnessed to enhance recovery after spinal cord injury (SCI). To do so, we developed a real-time closed-loop neuromodulation paradigm based on the synaptic eligibility trace to deliver VNS immediately after the most successful forelimb movements during motor rehabilitation. The strategy uses a control algorithm that adaptively scales stimulation threshold to trigger a brief 0.5 s train of VNS on trials in which pull forces fall within the top quintile of previous trials (Top 20% CLV; *Figure 1a,b*, and *Figure 1—figure supplement 1a*). To test the hypothesis that temporal precision is required for VNS-dependent effects, we employed a similar algorithm in which stimulation was delivered on the weakest quintile of trials (Bottom 20% CLV; *Figure 1b* and *Figure 1—figure supplement 1g*). Both algorithms deliver the same amount of VNS during rehabilitative training, but Bottom 20% CLV results in a significant delay between VNS and the most successful trials.

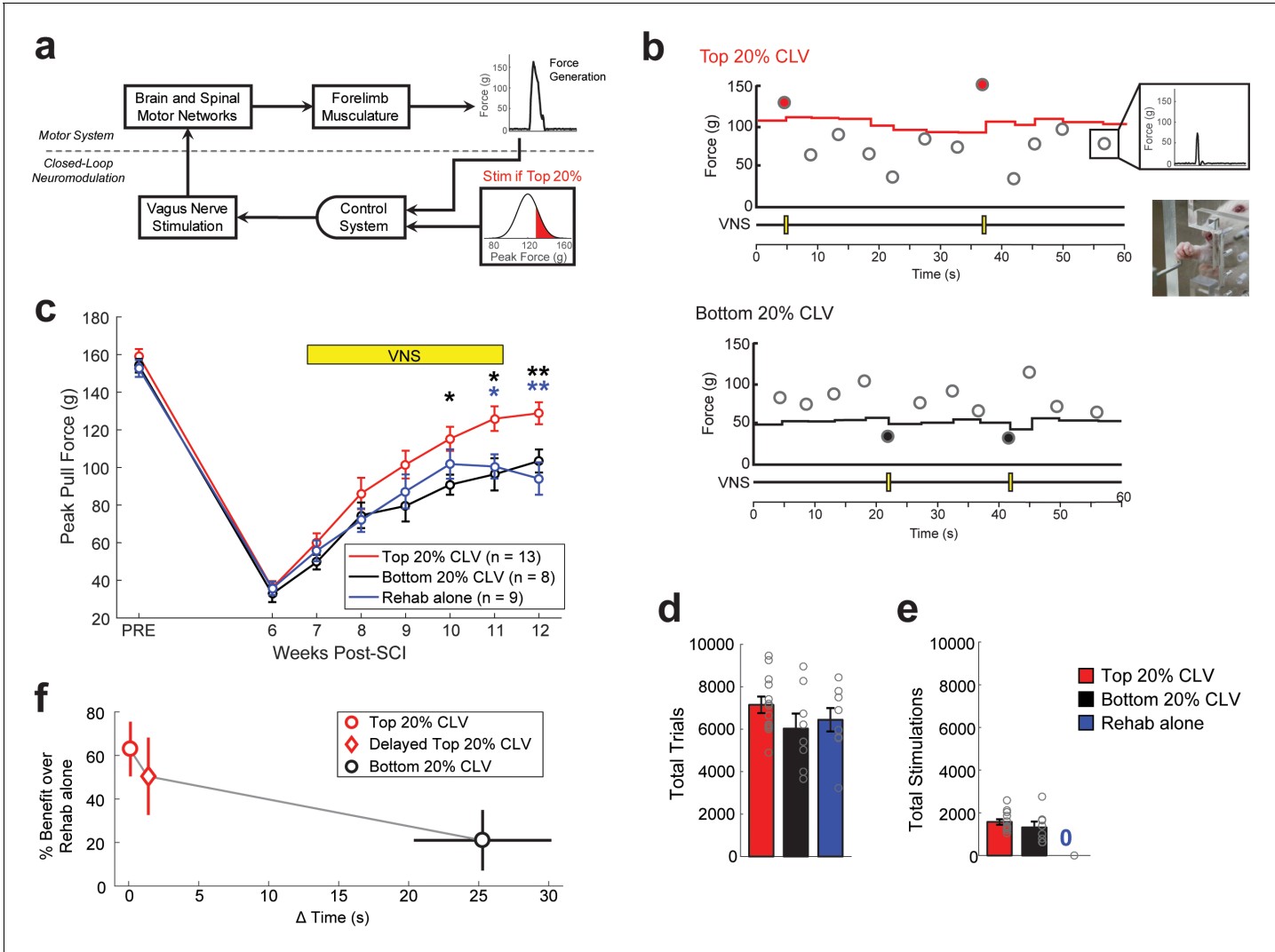

**Figure 1.** Precisely-timed closed-loop vagus nerve stimulation based on the synaptic eligibility trace enhances recovery after spinal cord injury. (**a**) Closed-loop neuromodulation to deliver vagus nerve stimulation to reinforce the most successful trials during rehabilitative training after SCI. (**b**) Top 20% CLV received a 0.5 s train of VNS on trials in which pull force falls within the highest quintile of previous pull forces. The Bottom 20% CLV group received VNS on trials in which pull force falls within the lowest quintile. Rehab alone performed equivalent rehabilitative training without VNS. Each circle represents peak pull force on an individual trial. Inset shows an animal performing the isometric pull task. See *Figure 1—figure supplement 1* for more detail. (**c**) Top 20% CLV significantly improves forelimb function after SCI compared to Bottom 20% CLV and Rehab alone, indicating that precisely-timed VNS enhances recovery. (**d,e**) Differences in the intensity of rehabilitative training or the amount of stimulations cannot account for improved recovery. A significant increase in recovery is observed with Top 20% CLV after correcting for number of trials and number of stimulations (ANCOVA, effect of group; number of trials: $F_{[1,1]}=11.89$, $p=0.0031$; number of stimulations: $F_{[1,1]}=9.57$, $p=0.0066$). Gray circles denote individual subjects. (**f**) CLV delivered within 2 s of successful trials increases recovery, whereas CLV separated 25 s from successful trials fails to yield substantial benefits. This time window is consistent with the synaptic eligibility trace hypothesis. Horizontal error bars for Top 20% CLV and Delayed Top 20% CLV are not visible because of their small size. In panel c, \*\*p<0.01, \*p<0.05 for t-tests across groups at each time point. The color of the asterisk denotes the group compared to Top 20% CLV. Error bars indicate S.E.M.

DOI: https://doi.org/10.7554/eLife.32058.003

The following figure supplements are available for figure 1:

**Figure supplement 1.** Adaptive Thresholding and Stimulation Timing.
DOI: https://doi.org/10.7554/eLife.32058.004

**Figure supplement 2.** Distribution of Pull Forces after SCI.
DOI: https://doi.org/10.7554/eLife.32058.005

# Results

To test whether CLV could improve recovery of motor function after SCI, rats were trained to perform an automated reach-and-grasp task measuring volitional forelimb strength (*Figure 1b*, *Video 1*) (*Hays et al., 2013*). Once proficient, rats received a right unilateral impact at spinal level C6 to impair function of the trained forelimb and underwent implantation of a bipolar cuff electrode on the left cervical vagus nerve (*Ganzer et al., 2016a*). SCI resulted in a 77% reduction in volitional forelimb strength, consistent with paresis observed in many cervical SCI patients (*Figure 1c*, PRE v. Wk 8, Paired t-test, t(29) = 37.34, p=4.4×10$^{-26}$, *Video 2*). Top 20% CLV substantially boosted recovery of volitional forelimb strength compared to equivalent rehabilitative training without CLV (Rehab alone), demonstrating that CLV enhances recovery of motor function after SCI (*Figure 1c*; Two-way repeated measures ANOVA, Interaction; F[6,120]=3.88, p=1.43×10$^{-3}$; *Videos 3–4*). CLV resulted in lasting recovery after the cessation of stimulation after week 11, consistent with the notion that CLV restores function in critical motor networks (Top 20% CLV; Wk 11 v. Wk 12; Paired t-test, t(12) = −0.89, p=0.38). Despite equivalent rehabilitation and a comparable number of stimulations delivered during task performance (*Figure 1d,e*), Bottom 20% CLV resulted in substantially diminished recovery compared to Top 20% CLV (*Figure 1c*, Two-way repeated measures ANOVA, Interaction; F [6,114]=2.40, p=0.03, *Video 5*) and failed to improve forelimb strength compared to Rehab alone. Together, these findings demonstrate that closed-loop neuromodulation paired with the most successful movements during rehabilitation improves recovery of motor function after cervical SCI.

The synaptic eligibility trace theory posits that neuromodulatory reinforcement must occur within seconds after neural activity to drive plasticity (*He et al., 2015*). To clarify how temporally precise CLV must be, a subset of rats received VNS delayed approximately 1.5 s after the top 50% most successful trials (Delayed Top 20% CLV, *Figure 1—figure supplement 1d*). This short delay resulted in comparable recovery to stimulation delivered immediately after a successful trial in the Top 20% CLV group (*Figure 1f*). Stimulation in the Bottom 20% CLV group was separated by 25 ± 5 s from the most successful trials and failed to drive substantial benefits (*Figure 1f*, *Figure 1—figure supplement 1g*). This absence of enhanced recovery despite delivery of CLV may be attributed to either the long delay or greater variance in the timing between stimulation and the most successful trials. These findings support a temporal precision limit for CLV near 10 s, consistent with the synaptic eligibility trace hypothesis (*He et al., 2015*).

To determine whether more pairings of VNS with successful trials would improve recovery, we utilized an adaptive algorithm in which VNS was delivered on at least the top 50% most successful trials, resulting in 2.5 times more stimulation pairings (Top 50% CLV, *Figure 1—figure supplement 1j*). Top 50% CLV substantially improved recovery of forelimb function compared to Rehab alone, which provides an independent confirmation that CLV enhances recovery after SCI (*Figure 2a*, Two-way repeated measures ANOVA, Interaction; F[6,174]=3.56, p=2.38×10$^{-3}$). The rate and degree of recovery were comparable in the Top 50% CLV and the Top 20% CLV groups (*Figure 2—figure supplement 1*), suggesting that timing is more important than quantity of stimulation.

Plasticity in remaining networks could be harnessed to support recovery after SCI (*Fink and Cafferty, 2016*; *Manohar et al., 2017*). Unilateral SCI resulted in extensive damage to gray matter, rubrospinal pathways, and propriospinal pathways in the right hemicord while largely sparing

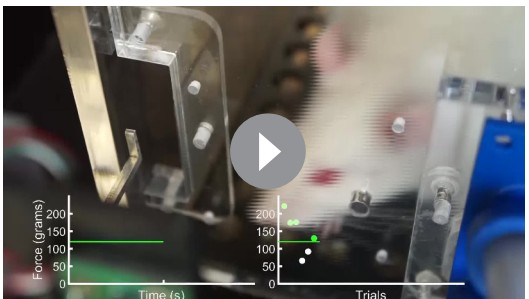

**Video 1.** Forelimb motor performance prior to SCI. Representative example of a rat performing the isometric pull task prior to lesion. Note that the rat is highly proficient and displays clear forelimb motor control. (A) The plot on the bottom left displays pull force applied to the handle on an individual trial. The green line indicates force threshold to trigger delivery of a reward pellet. Note that the threshold adaptively scales based on the median peak force of the ten antecedent trials. The triangle indicates peak pull force on a trial. (B) The plot on the bottom right shows performance over the course of the session. Each circle indicates peak force from an individual trial. Green circles indicate trials in which pull force exceeded the reward threshold. White circles indicate trials in which peak force did not exceed the reward threshold.
DOI: https://doi.org/10.7554/eLife.32058.006

the right dorsal corticospinal tract (CST) (*Figure 2b* and *Figure 2—figure supplement 2*). Thus, we used intracortical microstimulation to test the hypothesis that CLV enhances output from the cortico-spinal circuits to the impaired forelimb. CLV resulted in eight times more motor cortex sites that generated grasp movements in the impaired forelimb compared to Rehab alone (*Figure 2c* and *Figure 2—figure supplement 4*, Unpaired t-test, t(10) = 2.28, p=0.04), providing the first evidence that CLV induces large-scale plasticity in corticospinal networks after neurological injury.

We next tested the hypothesis that CLV improves recovery by increasing synaptic connections within the motor network controlling grasping muscles of the forelimb. We injected the retrograde transsynaptic tracer pseudorabies virus (PRV-152) into flexor digitorum profundus and palmaris longus and counted labeled neurons six days later. CLV resulted in a five-fold increase in labeled neurons in motor cortex compared to Rehab alone (*Figure 2d* and *Figure 2—figure supplement 5*, Unpaired t-test, t(9) = 7.63, p=3.2×10$^{-5}$). The magnitude of this increase in synaptic connectivity is comparable to the seven-fold increase in the number of motor cortex sites that produce grasp. CLV failed to increase neuronal labeling of spinal motor neurons, red nucleus neurons or propriospinal neurons (*Figure 2d*). Additionally, CLV did not influence lesion extent (*Figure 2e*, *Figure 2—figure supplement 2*). Together, these results are consistent with anatomical plasticity in the spared corticospinal network contributing to enhanced recovery when CLV is added to rehabilitative training after SCI (*Figure 3*).

The observation that CLV improves recovery and enhances functional and anatomical plasticity in corticospinal networks suggests that CLV may prove ineffective if the CST is destroyed. Given the severity and anatomical heterogeneity of damage observed in SCI patients (*Sekhon and Fehlings, 2001*), such a finding would limit the clinical utility of CLV. We therefore evaluated motor recovery in a bilateral injury model that virtually eliminates the CST on both sides of the cord (*Figure 4b*). Despite profound damage, CLV more than doubled the degree of forelimb motor recovery compared to Rehab alone (*Figure 4a*, Two-way repeated measures ANOVA, Interaction; F[6,144]=5.29, p=7.62×10$^{-5}$). The observation that CLV can improve recovery following bilateral SCI suggests CLV could be clinically useful. We hypothesized that CLV enhances recovery by promoting plasticity in the rubrospinal and propriospinal pathways, which were damaged, but not eliminated, by this injury (*Figure 4b* and *Figure 4—figure supplement 1*). Indeed, CLV doubled the number of labeled red nucleus neurons and C3/4 propriospinal neurons compared to Rehab alone (*Figure 3d* and *Figure 4—figure supplement 3*, Unpaired t-test, Red Nucleus: t(4) = 3.89, p=0.018; Propriospinal: t(4) = 2.77, p=0.05). Consistent with the extensive damage to the corticospinal pathway, CLV had no effect on reorganization of motor cortex (*Figure 4c* and *Figure 4—figure supplement 4*, Unpaired t-test, t(12) = −0.13, p=0.90) and failed to increase the number of labeled neurons in the motor cortex (*Figure 3d*, Unpaired t-test, t(4) = 0.83, p=0.45). These results suggest that CLV is capable of supporting recovery following SCI by strengthening anatomical connectivity within remaining pathways (*Figure 4—figure supplement 5*).

## Discussion

In this study, we developed a novel closed-loop neuromodulation strategy to make use of the high temporal precision of the synaptic eligibility trace. We demonstrate that activation of the vagus nerve improves recovery when reliably delivered within seconds of a successful movement, and we provide the first evidence that CLV enhances reorganization of synaptic connectivity in remaining networks in two non-overlapping models of SCI. The flexibility to promote reorganization in a range of pathways is a critical benefit of CLV, given the great heterogeneity in the etiology, location, and extent of damage present in SCI patients.

Classical studies by Skinner demonstrate that adaptive reinforcement of successive approximations, or shaping, drives behavior toward a desired response (*Skinner, 1953*). This principle has been adopted for use in rehabilitation, with the intention to reinforce successively better movements (*Wood, 1990*). We made use of this concept by applying an adaptively-scaled stimulation threshold to deliver CLV with the most successful forelimb movements during rehabilitation. Enhanced recovery was observed only when CLV was paired with trials that approximated the desired outcome, highlight the importance of timing for closed-loop stimulation to shape behavioral outcomes and maximize recovery.

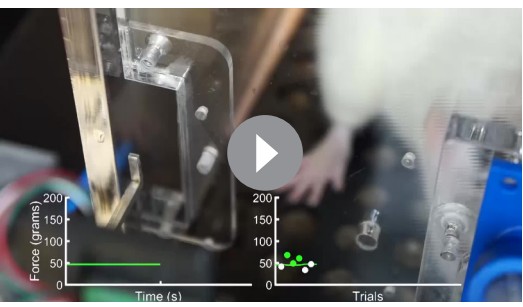

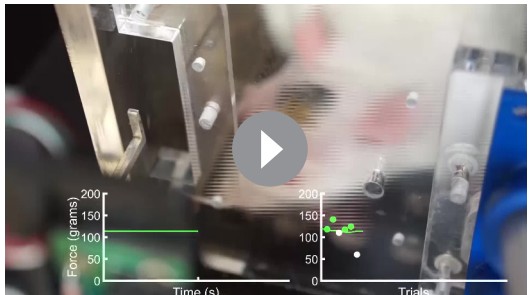

**Video 2.** Forelimb motor performance after unilateral SCI. Representative example of a rat performing the isometric pull task four weeks after unilateral SCI. The rat displays a notable reduction in peak pull force, consistent with forelimb paresis, and clear deficits in motor control. (A) The plot on the bottom left displays pull force applied to the handle on an individual trial. The green line indicates the adaptive-scaled force threshold to trigger delivery of a reward pellet. Note that the threshold adaptively scales based on the median peak force of the ten antecedent trials. The triangle indicates peak pull force on a trial. (B) The plot on the bottom right shows performance over the course of the session. Each circle indicates peak force from an individual trial. Green circles indicate trials in which pull force exceeded the reward threshold. White circles indicate trials in which peak force did not exceed the reward threshold.

DOI: https://doi.org/10.7554/eLife.32058.007

**Video 3.** Performance after Rehabilitation Alone. Representative example of a unilateral SCI rat performing the isometric pull task at the conclusion of six weeks of rehabilitative training alone. Note the sustained impairments in forelimb pull force generation and motor control despite intensive rehabilitative training. (A) The plot on the bottom left displays pull force applied to the handle on an individual trial. The green line indicates the adaptive-scaled force threshold to trigger delivery of a reward pellet. Note that the threshold adaptively scales based on the median peak force of the ten antecedent trials. The triangle indicates peak pull force on a trial. (B) The plot on the bottom right shows performance over the course of the session. Each circle indicates peak force from an individual trial. Green circles indicate trials in which pull force exceeded the reward threshold. White circles indicate trials in which peak force did not exceed the reward threshold.

DOI: https://doi.org/10.7554/eLife.32058.008

The magnitude of neuromodulatory activation elicited by an event is directly proportional to the surprise, or unpredictability, of the event (*Hangya et al., 2015*; *Hollerman and Schultz, 1998*; *Sara and Segal, 1991*). This phenomenon is ascribed to reward prediction error (*Schultz, 2002*). Unsurprising events fail to activate neuromodulatory systems, and even rewarding events fail to trigger neuromodulator release if they are expected. We posit the predictability and accompanying tedium of long, frustrating rehabilitation and the minimal reinforcement of practicing a previously simple motor task blunts plasticity and limits recovery after SCI. The closed-loop neuromodulation strategy developed here circumvents this by artificially engaging neuromodulatory networks and providing a repeated, non-adapting reinforcing signal typically associated with surprising consequences (*Hays, 2016*; *Hulsey et al., 2017*; *Hulsey et al., 2016*). CLV drives temporally-precise neuromodulatory release to convert the synaptic eligibility trace in neuronal networks that generate optimal motor control to long-lasting plasticity (*He et al., 2015*).

CLV is a minimally-invasive, safe strategy to provide precisely-timed engagement of multiple neuromodulatory networks to boost plasticity during rehabilitation (*Hays, 2016*). Preliminary results in chronic stroke and tinnitus patients highlight the clinical potential of CLV, while delivering less than 1% of the total FDA-approved amount of stimulation (*Dawson et al., 2016*; *De Ridder et al., 2014*; *Ben-Menachem, 2001*). Moreover, the flexibility to deliver stimulation with a variety of rehabilitative exercises raises the possibility to design CLV-based to target motor dysfunction of the lower limbs, somatosensory loss, and bowel and bladder issues, all of which are prevalent in SCI patients. Delineation of the timing requirements and documentation of neuronal changes driven by CLV in this study provide a framework for development of this strategy for a range of neurological conditions, including stroke, peripheral nerve injury, and post-traumatic stress disorder (*Hays, 2016*; *Lozano, 2011*).

# Materials and methods

## Key resources table

| Reagent type (species) or resource | Designation | Source or reference | Identifiers |
| --- | --- | --- | --- |
| Pseudorabies Virus Bartha Strain containing the CMV-EGFP reporter cassette | PRV-152 | NIH Center for Neuroanatomy with Neurotropic Viruses (CNNT) | PRV-152 |

## Experimental Design

All procedures performed in the study were approved by the University of Texas at Dallas Institutional Animal Care and Use Committee (Protocols: 14–10 and 99–06). Adult female Sprague Dawley rats (N = 181) used in this study were housed one per cage (12 hr light/dark cycle). Twelve experimentally naïve rats were used for control experiments. One hundred and sixty-nine rats were trained to proficiency on the isometric pull task as in our previous studies (*Khodaparast et al., 2016*; *Pruitt et al., 2016*; *Hays et al., 2013*; *Sloan et al., 2015*; *Hays et al., 2014b*; *Hays et al., 2016*; *Khodaparast et al., 2013*; *Pruitt et al., 2014*; *Meyers et al., 2017*). Sample sizes were based estimated effect size determined in our initial pilot studies and are consistent with comparable previous studies. Trained rats were food restricted Monday-Friday to provide task motivation (*ad libitum* access to water). Because of the cage geometry, only the right forelimb can be used to reach the pull handle to trigger a food reward. After reaching task proficiency (85% success rate on ten consecutive sessions), rats received a unilateral contusive injury (N = 128) or bilateral contusive injury (N = 41) of the cervical spinal cord. After recovery, rats received a vagus nerve cuff electrode and resumed training on the isometric pull task. In addition to the food reward, rats were dynamically allocated to balanced groups to receive a brief burst of vagus nerve stimulation (VNS) on appropriate trials. Rehabilitative training, consisting of freely performing the task, continued for six weeks. No VNS was delivered in any group on the final week of rehabilitative training, to allow assessment of lasting effects of stimulation. Terminal motor cortex mapping or transsynaptic tracing experiments occurred the week following the end of therapy in a subset of unilateral (N = 23) and bilateral SCI rats (N = 20). Eighty-seven rats were excluded from the study due to mortality (N = 20), inability to perform the task after injury (N = 25), or VNS device failure (N = 42). Device failure included mechanical failure of the headmount or loss of stimulation efficacy, determined by a cuff impedance >25 kΩ or by the absence of a reduction in blood oxygenation in response to a train of VNS while under anesthesia (described below). This is a standard method to evaluate VNS efficacy (*Loerwald et al., 2017*; *Borland et al., 2018*). Animals that failed to demonstrate a reliable drop in oxygen saturation at the end of therapy were excluded. Bilateral SCI rats were given two additional weeks of recovery time due to their larger spinal lesion and slower return to recumbency (*Figure 4—figure supplement 6*). Other than therapy start time (6 vs. 8 weeks post-SCI), all training and assessment was identical for unilateral and bilateral SCI rats. All source data indexed across animals can be found in *Supplementary file 1–4*.

## Volitional forelimb Force Generation Assessment

The isometric pull task is a fully automated and quantitative assay to measure multiple parameters of forelimb force generation and was performed similar to previous descriptions (*Khodaparast et al., 2016*; *Pruitt et al., 2016*; *Hays et al., 2013*; *Sloan et al., 2015*; *Hays et al., 2014b*, *2016*; *Khodaparast et al., 2013*; *Pruitt et al., 2014*; *Meyers et al., 2017*). Isometric pull training sessions consisted of two 30 min sessions (separated by at least 2 hr) five days per week. Experimenters were blind to treatment group at all times throughout behavioral testing. Early in training, rats were encouraged to interact with the pull handle by dispensing pellets (45 mg chocolate-flavored pellets, Bio-Serv; Flemington, NJ) when they approached or touched the lever. The pull handle was initially located inside the test chamber and then slowly retracted outside of the behavioral chamber to encourage reaching with the right paw. A trial was initiated when the rats exerted at least 10 g of force on the pull handle. A trial window of 2 s started after trial initiation where the animal could receive a reward by pulling with a force exceeding a reward threshold. The reward threshold was scaled adaptively based on the median peak force of the 10 preceding trials, with a fixed bounded minimum of 10 grams and maximum of 120 g based on previous studies (*Figure 1—figure*

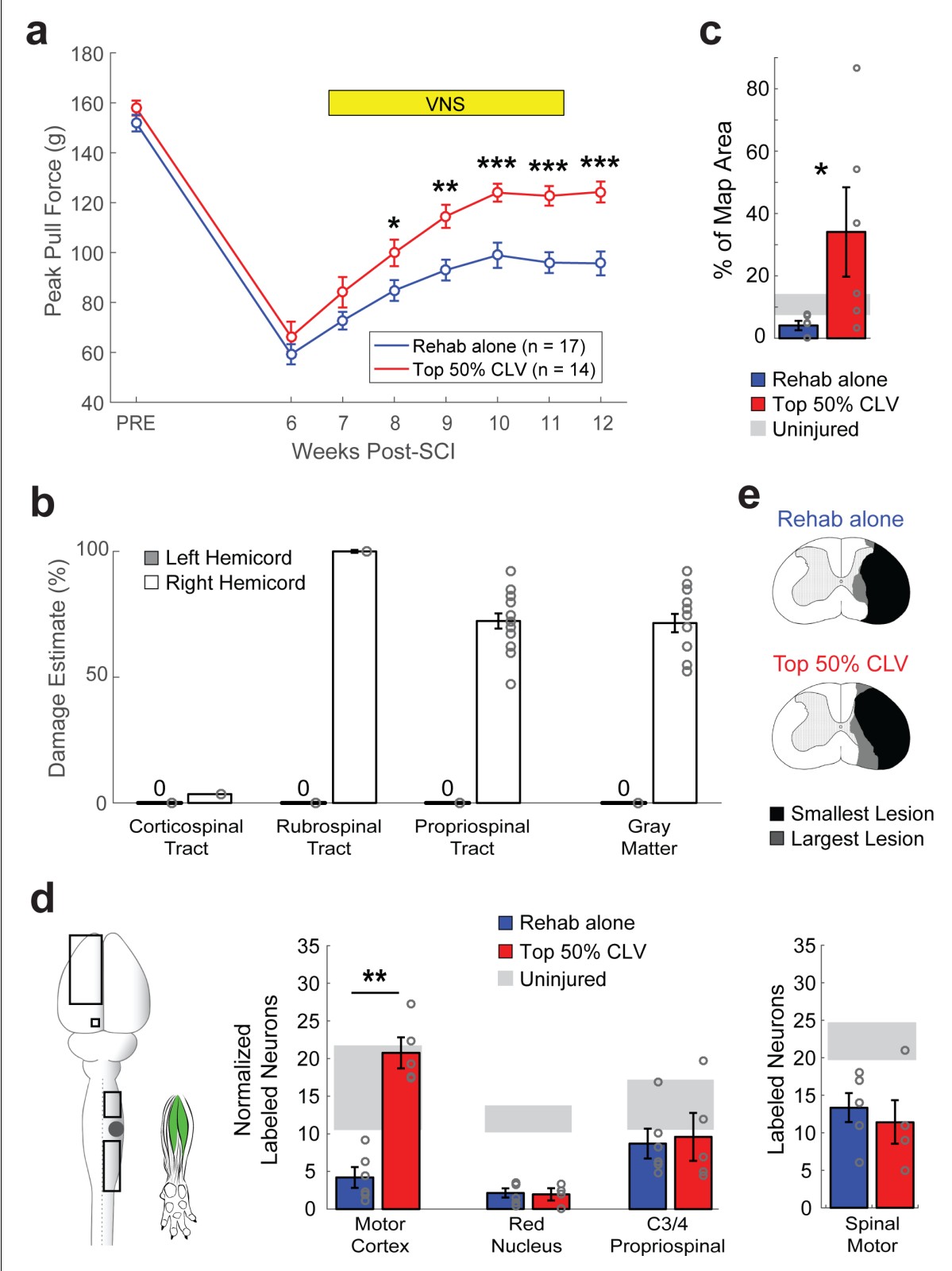

**Figure 2.** CLV enhances plasticity in spared corticospinal networks and improves functional recovery after unilateral SCI. (a) Top 50% CLV significantly improved recovery of forelimb function compared to Rehab alone. Sustained recovery was observed on week 12 after the cessation of stimulation, indicating lasting benefits. (b) Unilateral SCI caused substantial damage to gray matter, rubrospinal, and propriospinal tracts in the right hemicord, while largely sparing the right corticospinal tract and the entirety of the left hemicord. (c) ICMS reveals that Top 50% CLV significantly increases the area

*Figure 2 continued on next page*

*Figure 2 continued*

of the forelimb motor cortex evoking rehabilitated grasping movements compared to Rehab alone (N = 6,6). (d) Retrograde transneuronal tracing with PRV-152 was performed to evaluated anatomical connectivity from the left motor cortex neurons, left red nucleus neurons, and right C3/4 propriospinal neurons to grasping muscles in the trained (right) forelimb. Top 50% CLV restores connectivity and results in a significant increase in labeled neurons in the motor cortex compared to Rehab alone (N = 5,6). No changes were observed in red nucleus or C3/4 propriospinal neurons. Black boxes indicate ROIs; gray dot indicates lesion epicenter; inset shows injected muscles. (e) CLV does not affect lesion size. In all panels, gray circles denote individual subjects. In all panels, ***p<0.001, **p<0.01, *p<0.05 for t-tests across groups. Error bars indicate S.E.M.

DOI: https://doi.org/10.7554/eLife.32058.011

The following figure supplements are available for figure 2:

**Figure supplement 1.** Top 20% CLV and Top 50% CLV display comparable recovery.

DOI: https://doi.org/10.7554/eLife.32058.012

**Figure supplement 2.** CLV does not affect lesion size after Unilateral SCI.

DOI: https://doi.org/10.7554/eLife.32058.013

**Figure supplement 3.** CLV does not influence the number of trials performed during rehabilitative training.

DOI: https://doi.org/10.7554/eLife.32058.014

**Figure supplement 4.** Motor Cortex Movement Representations.

DOI: https://doi.org/10.7554/eLife.32058.015

**Figure supplement 5.** Distribution of eGFP+ neurons with Rehabilitation alone or Top 50% CLV after Unilateral SCI.

DOI: https://doi.org/10.7554/eLife.32058.016

supplement 1) (*Ganzer et al., 2016a*; *Meyers et al., 2017*). Thus, rats received rewards on trials that exceeded either the median peak force from the previous 10 trials or 120 g. The reward threshold was set to 10 g for the first 10 trials of a training session and adaptive scaled for the remaining trials (*Figure 1—figure supplement 1*). This reward threshold paradigm was used for all groups at all timepoints during the study.

Rats were trained until they reached proficiency, defined as a 10 consecutive sessions in which greater than 85% of trials exceeded 120 g. After reaching isometric pull task proficiency, rats were given a cervical unilateral or bilateral SCI at spinal level C6. Post-injury baseline force generation assessment occurred on week six for unilateral SCI and week eight for bilateral contusion SCI and consisted of 2 × 30 min sessions per day across two consecutive days (POST; *Figures 1C* and *2A,* and *3A*). Random group assignment was used to determine which rats received VNS for the first 75% of group assignment decisions. To ensure well-balanced treatment groups, the final 25% of rats were assigned to groups based on their post-injury performance. Rehabilitative training continued for 6 weeks with VNS delivered when appropriate. MotoTrak Software (Vulintus, Inc.) was used to record and display experimental data during the performance of the isometric pull task similar to previous studies (*Hays et al., 2013*; *Ganzer et al., 2016a*; *Sloan et al., 2015*; *Pruitt et al., 2014*; *Meyers et al., 2016*). A microcontroller board (Vulintus, Inc.; Dallas Texas USA) sampled the force transducer every 10 ms and relayed information to the MotoTrak software for offline analysis. For rats receiving VNS, stimulation was triggered by the behavioral software on appropriate trails during rehabilitative training. Peak pull force (maximum force generated in a trial, g) was calculated for every rat for every week of behavior. A Two-way repeated measures ANOVA was used to compared peak pull forces in each treatment condition across time, followed by *post hoc* Bonferroni-corrected unpaired t-tests where appropriate (*Figures 1C*, *2A* and *4A*). Percent benefit over Rehab alone was calculated as the recovery of peak pull force after therapy normalized to the average recovery observed in the Rehab alone group (*Figure 1J*). The distribution of pull forces after injury is shown in *Figure 1—figure supplement 2*. Behavioral data for each week for all individual subjects is available in *Supplementary file 1*.

## Cervical Spinal Cord Injury (SCI) Surgery

Experimenters were blind to the group of the rat during surgery. All surgeries were performed using aseptic technique under general anesthesia. Rats were anesthetized with ketamine (50 mg/kg), xylazine (20 mg/kg), and acepromazine (5 mg/kg) for all procedures (i.p.). Heart rate and blood oxygenation was monitored during surgery. After achieving isometric pull task proficiency, rats received either a right side (unilateral) or midline (bilateral) C6 spinal cord contusive impact using surgical technique from previous studies (*Ganzer et al., 2016a*). A right side or bilateral dorsal C5

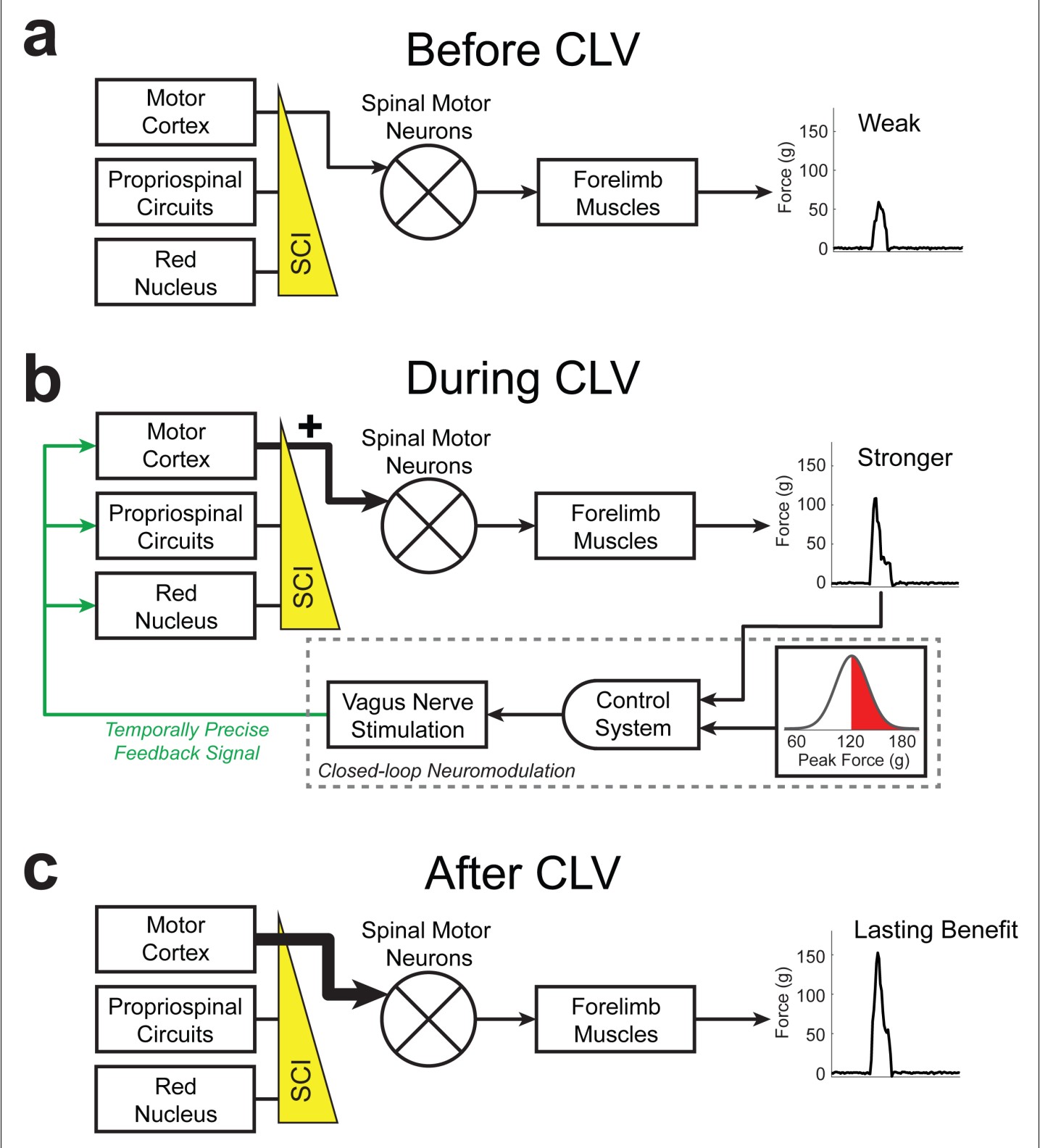

**Figure 3.** Schematic of CLV-dependent recovery after unilateral SCI. (**a**) After unilateral SCI, loss of motor out from rubrospinal and propriospinal networks results in forelimb paresis and impairments in motor control. (**b**) The addition of CLV provides temporally-precise feedback on the most successful trials to facilitate training-dependent plasticity in remaining motor networks. (**c**) The benefits of CLV persist after the cessation of closed-loop stimulation because connectivity in critical neural circuits has been restored.

DOI: https://doi.org/10.7554/eLife.32058.017

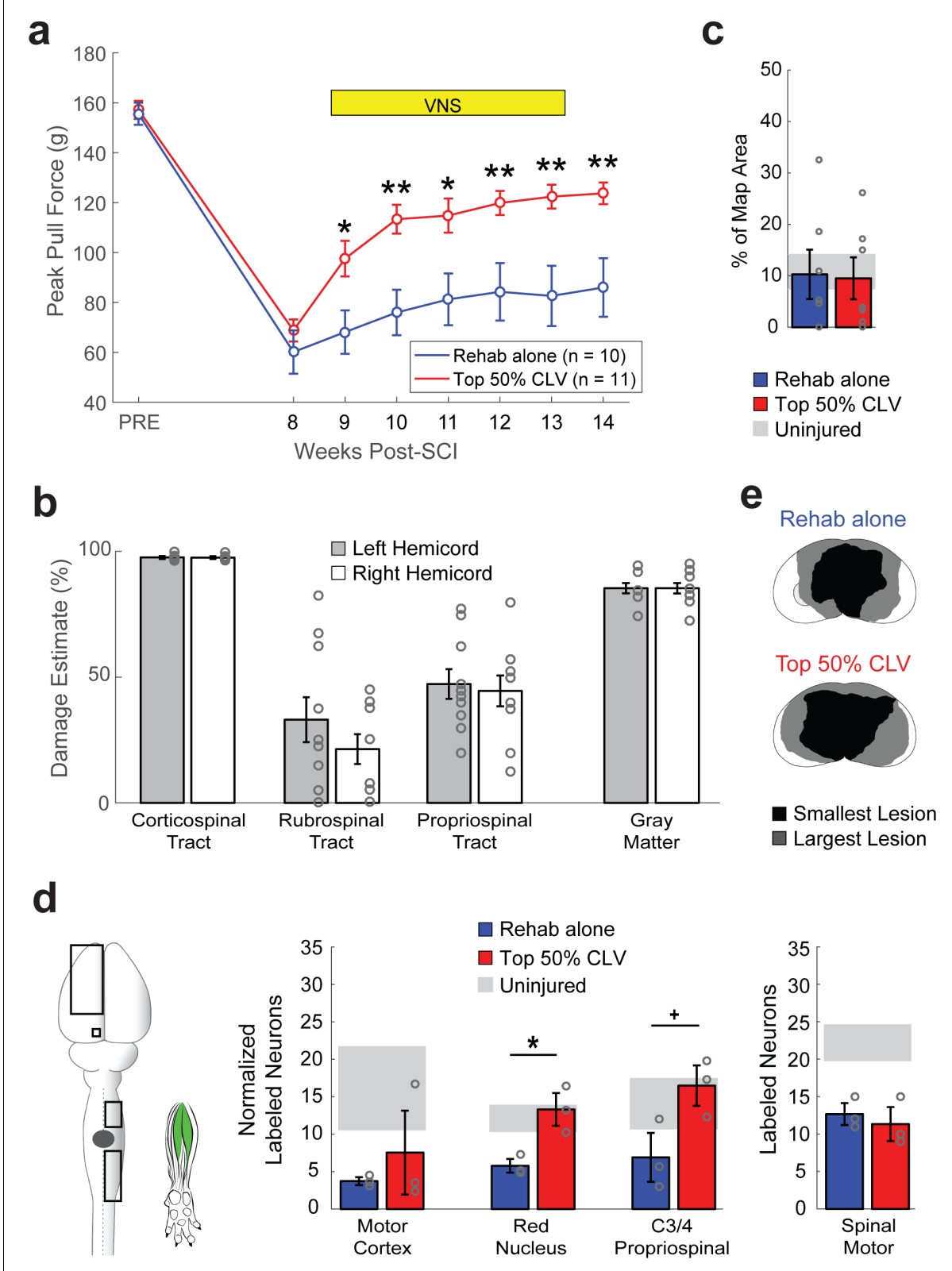

**Figure 4.** CLV enhances synaptic plasticity and recovery after bilateral SCI. (a) After bilateral SCI, Top 50% CLV substantially enhanced recovery of volitional forelimb strength compared to Rehab alone. Improved function was maintained on week 14 after the cessation of CLV, indicative of lasting recovery. (b) Bilateral SCI resulted in virtually complete bilateral ablation of the corticospinal tract and substantial damage to gray matter. The rubrospinal and propriospinal tracts were lesioned, but partially remaining. (c) Unlike after unilateral SCI, Top 50% CLV failed to increase the area of the

*Figure 4 continued*

motor cortex evoking rehabilitated grasping movements compared to Rehab alone (N = 7,7). (**d**) CLV significantly increased synaptic connectivity from the left red nucleus neurons and right C3/4 propriospinal neurons to grasping muscles compared to Rehab alone (N = 3,3). Black boxes indicate ROIs; gray dot indicates lesion epicenter; inset shows injected muscles. In all panels, gray circles denote individual subjects. In all panels, **p<0.01, *p<0.05, +P = 0.05 for t-tests across groups. Error bars indicate S.E.M.
DOI: https://doi.org/10.7554/eLife.32058.018

The following figure supplements are available for figure 4:

**Figure supplement 1.** CLV does not affect lesion size after Bilateral SCI.
DOI: https://doi.org/10.7554/eLife.32058.019

**Figure supplement 2.** CLV does not influence the number of trials performed during rehabilitative training after bilateral SCI.
DOI: https://doi.org/10.7554/eLife.32058.020

**Figure supplement 3.** Distribution of eGFP+ neurons with Rehabilitation alone or Top 50% CLV after Bilateral SCI.
DOI: https://doi.org/10.7554/eLife.32058.021

**Figure supplement 4.** Motor Cortex Movement Representations.
DOI: https://doi.org/10.7554/eLife.32058.022

**Figure supplement 5.** CLV drives plasticity in remaining motor networks to support recovery after bilateral SCI.
DOI: https://doi.org/10.7554/eLife.32058.023

**Figure supplement 6.** Post-SCI Time until Recumbency and Paw Placement.
DOI: https://doi.org/10.7554/eLife.32058.024

laminectomy was performed for rats receiving a unilateral or bilateral SCI, respectively. The vertebral column was stabilized using spinal microforceps. For unilateral SCI, the right spinal hemicord was contused using the Infinite Horizon Impact Device with a force of 200 kilodynes and zero dwell time as previously reported (Precision Systems and Instrumentation, Lexington, KY; impactor tip diameter = 1.25 mm) (*Ganzer et al., 2016a*). For bilateral SCI rats, the midline of the spinal cord was contused with a force of 225 kilodynes and zero dwell time (impactor tip diameter = 2.5 mm). The skin overlying the exposed vertebrae was then closed in layers and the incised skin closed using surgical staples. All rats received buprenorphine (s.c., 0.03 mg/kg, 1 day post-op), enrofloxacin (s.c., 10 mg/kg, 3 days post-op) and Ringer's solution (s.c., 10 mL, 3 days post-op) immediately after surgery and continuing post-operatively. All rats were monitored daily for at least 1 week post-injury. We documented time to return to recumbency, defined as the return of the righting reflex and ability to self-feed, and plantar placement following SCI. After bilateral SCI, rats took significantly longer to return to recumbency and forepaw plantar placement compared to unilateral SCI rats (*Figure 4—figure supplement 6*). Therefore, bilateral SCI rats started therapy 2 weeks later. After injury, rats were hand fed twice daily and given Ringer's solution (s.c., 10 mL) for up to 1 week post-injury to maintain a healthy diet.

## Vagus nerve stimulation cuff implantation surgery

A two-channel connector headmount and vagus nerve stimulating cuff were implanted on post-injury week six for unilateral and week eight for bilateral SCI rats similar to previous studies (*Engineer et al., 2011*; *Khodaparast et al., 2016*; *Pruitt et al., 2016*; *Hays et al., 2014a*; *Hays et al., 2014b*; *Hays et al., 2016*; *Khodaparast et al., 2013*; *Khodaparast et al., 2014*; *Borland et al., 2016*). Regardless of group assignment, all rats underwent implantation of the headmount and cuff. Stimulation of the left cervical branch of the vagus nerve was performed using low current levels to avoid cardiac effects (*Engineer et al., 2011*). Incised skin was closed using suture. All rats received enrofloxacin (s.c., 10 mg/kg) following surgery and as needed at the sign of infection. To confirm cuff functionality and proper placement, heart rate, respiration, and blood oxygenation saturation during VNS (0.8 mA, 30 Hz, 100 µs pulse width, 1–5 s train duration) were monitored under anesthesia via pulse oximetry after cuff implant and at the end of therapy. Animals that failed to demonstrate a reliable drop in oxygen saturation at the end of therapy were excluded. Stimulation under anesthesia briefly suppressed cardiopulmonary function and was not more severe or lower threshold in SCI rats compared to intact rats.

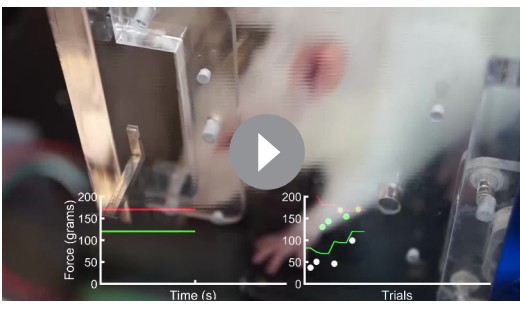

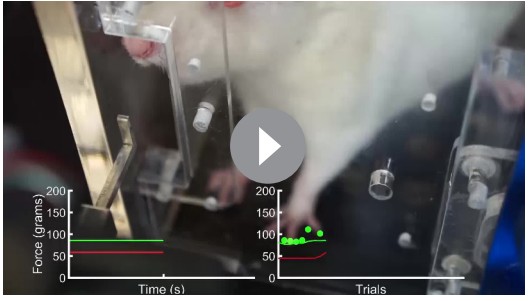

**Video 4.** Performance after Top 20% CLV Representative example of a unilateral SCI rat performing the isometric pull task at the conclusion of six weeks of Top 20% CLV therapy. Note the improvements in motor control and increase in forelimb pull force compared to after SCI. (A) The plot on the bottom left displays pull force applied to the handle on an individual trial. The green line indicates the adaptive-scaled force threshold to trigger delivery of a reward pellet. Note that the threshold adaptively scales based on the median peak force of the ten antecedent trials. The red line denotes the stimulation threshold based on the top 20% of peak pull force from the preceding 10 trials. Vertical red lines indicate VNS delivery when pull force exceeds the stimulation threshold. The triangle indicates peak pull force on a trial. (B) The plot on the bottom right shows performance over the course of the session. Each circle indicates peak force from an individual trial. Green circles indicate trials in which pull force exceeded the reward threshold. Green circles with a red border indicate trials on which VNS was delivered (the top 20% of trials). White circles indicate trials in which peak force did not exceed the reward threshold.
DOI: https://doi.org/10.7554/eLife.32058.009

**Video 5.** Performance after Bottom 20% CLV Representative example of a unilateral SCI rat performing the isometric pull task at the conclusion of six weeks of Bottom 20% CLV therapy. Note the remaining deficits in forelimb weakness and reduced motor control compared to Top 20% CLV. (A) The plot on the bottom left displays pull force applied to the handle on an individual trial. The dashed line indicates the adaptive-scaled force threshold to trigger delivery of a reward pellet. Note that the threshold adaptively scales based on the median peak force of the ten antecedent trials. The red line denotes the stimulation threshold based on the bottom 20% of peak pull force from the preceding 10 trials. Vertical red lines indicate VNS delivery when pull force fails to exceed the stimulation threshold. The triangle indicates peak pull force on a trial. (B) The plot on the bottom right shows performance over the course of the session. Each circle indicates peak force from an individual trial. Green circles indicate trials in which pull force exceeded the reward threshold. White circles indicate trials in which peak force did not exceed the reward threshold. White circles with a red border indicate trials on which VNS was delivered (the bottom 20% of trials).
DOI: https://doi.org/10.7554/eLife.32058.010

## Vagus nerve stimulation parameters

VNS was triggered by the behavioral software during rehabilitative training based on the stimulation threshold for each group, similar to previous studies (*Khodaparast et al., 2016*; *Pruitt et al., 2016*; *Hays et al., 2014b*, *2016*; *Khodaparast et al., 2013*). Each stimulation train consisted of 16 × 100 μsec 0.8 mA biphasic pulses delivered at 30 Hz. An adaptive stimulation threshold specific to each CLV group was used to determine stimulation delivery during rehabilitative training (*Figure 1—figure supplement 1*). The stimulation threshold was adaptively scaled based on the 10 antecedent trials, with each group receiving VNS triggered on trials which fall into the appropriate range. Rats in the Top 20% CLV group (N = 13) received VNS on trials in which pull force exceeded the top quintile of the previous ten trials, with no minimum or maximum. In the majority of these subjects (N = 9), VNS was delivered immediately (~50 msec) after pull force exceeded the stimulation threshold (*Figure 1—figure supplement 1A*). No stimulation was delivered on the first 10 trials during a training session. In a different subset of subjects (Delayed Top 20% CLV, N = 4), VNS was delivered at the end of the 2 s trial window on trials in which exceeded the stimulation threshold, independent of the when the threshold was crossed (*Figure 1—figure supplement 1D*). These groups displayed comparable performance (Top 20% CLV v. Top 20% CLV Delay, Week 12, Unpaired t-test, p=0.78) and were thus combined for analysis in *Figure 1C–E*. Rats in the Bottom 20% CLV group (N = 8) received VNS on trials in which pull force failed to exceed the bottom quintile of the previous ten trials, with no minimum or maximum. VNS was delivered at the end of the 2 s trial window if pull force was below the threshold (*Figure 1—figure supplement 1G*). Rats in the Top 50% CLV group received

VNS on trials that exceeded the median pull force of the previous 10 trials or exceeded 120 g. VNS was delivered immediately (~50 msec) after pull force exceeded the stimulation threshold *Figure 1—figure supplement 1J*). No groups received VNS on the final week of rehabilitative training (Week 12 for unilateral and Week 14 for bilateral SCI) to assess effects lasting after the cessation of stimulation. These parameters do not cause discomfort and do not alter reaching behavior (*Hulsey et al., 2016*; *Porter et al., 2012*).

## Intracortical Microstimulation Mapping

Terminal intracortical microstimulation mapping (ICMS) of motor cortex was performed in a subset of unilateral SCI (Rehab alone, N = 6; Top 50% CLV, N = 6) and bilateral SCI (Rehab alone, N = 7; Top 50% CLV, N = 7) rats at the end of therapy. A group of uninjured rats were used for control (Naïve, N = 7). Rats were anesthetized with and injection (i.p.) of ketamine (50 mg/kg), xylazine (20 mg/kg), and acepromazine (5 mg/kg). A cisternal drain was performed to reduce ventricular pressure and cortical edema during mapping (*Hulsey et al., 2016*; *Porter et al., 2012*). A craniotomy was then performed to expose left motor cortex. Intracortical microstimulation (ICMS) was delivered in motor cortex at a depth of 1.75 mm using a low impedance tungsten microelectrode with an interpenetration resolution of 500 µm (100 kΩ – 1 MΩ electrode impedance; FHC Inc., Bowdin, MD; biphasic ICMS at 333 Hz, 50 ms train duration, 200 µsec pulse width, 0–200 µA current). Mapping experiments were performed blinded with two experimenters similar to previous studies (*Hulsey et al., 2016*; *Porter et al., 2012*). The first experimenter positioned the electrode for ICMS and recorded movement data. The second experimenter, blind to the experimental group of the animal and electrode position, delivered ICMS and classified movements. Movement threshold was first defined. ICMS current was then increased by no more than 50% to facilitate movement classification using visual inspection. Movements were classified into the following categories similar to previous studies: vibrissae, neck, jaw, digit, wrist, elbow, shoulder, hindlimb and trunk (*Brown and Teskey, 2014*; *Ganzer et al., 2016b*). The cortical area (mm [*Levy et al., 2016*]) and movement threshold (µA) for each movement category was calculated for each group (*Figure 2—figure supplement 4* and *Figure 4—figure supplement 4*). Based on the 500 µm inter-electrode spacing, each stimulation site eliciting a movement was counted as 0.25 mm$^2$. Movement area and threshold was assessed using One-way ANOVA and unpaired t-tests. Data for all movement classifications for each subject is available in *Supplementary file 2*.

## Pseudorabies virus retrograde transneuronal tracing

Transsynaptic tracer injections using pseudorabies virus 152 (PRV-152) were performed in a subset of unilateral SCI (Rehab, n = 6; VNS + Rehab, n = 5) and bilateral SCI (Rehab, n = 3; VNS + Rehab, n = 3) rats following the respective end of therapy. A group of uninjured rats were used for control (Naïve, n = 5). PRV-152 was a generous gift from the lab of Dr. Lynn Enquist and colleagues at Princeton University and was grown using standard procedures (*Card and Enquist, 2014*). An incision was made over the medial face of the radius and ulna of the trained limb to expose the forelimb grasping muscles flexor digitorum profundus and palmaris longus. 15 µL of PRV-152 (~8.06 ± 0.49 x 10$^8$ plaque-forming units) was injected into the belly of each muscle across three separate sites. The incision was then closed with non-absorbable suture. We conducted detailed pilot studies to determine the optimal time of viral infection to allow for layer five cortical labeling. At 5–5.5 days post-infection we observed little to no cortical labeling for injured or uninjured animals. At 6–6.5 days post-infection we observed consistent layer five cortical labeling across injured or uninjured animals. Therefore, 6–6.5 days was used as our PRV-152 infection duration for our transsynaptic tracing studies. Rats were anesthetized with sodium pentobarbital (50 mg/kg, i.p.) and transcardially perfused with 4% paraformaldehyde in 0.1 M PBS (pH 7.5) at 6–6.5 days after injection. The brain and spinal cord were removed. Spinal roots were kept for anatomical reference. Tissue was then post-fixed overnight and cryoprotected in 30% sucrose.

Quantification was limited to the spinal motor neurons, C3/4 cervical propriospinal neurons, red nucleus neurons, and cortical layer five neurons because these regions exhibited consistent labeling and were specifically related to our hypotheses. The whole neuraxis from the rostral tip of the forebrain to spinal level T3 was blocked and frozen at −80 C in Shandon M1 embedding matrix (Thermo Fisher Scientific; Waltham, MA). Coronal forebrain and midbrain sections were sliced and slide-

mounted at 35 µm using a cryostat (from the rostral tip of forebrain to 13 mm caudal). Coronal spinal cord sections were sliced and slide mounted at 50 µm (from C4 – T3). After coverslipping, slides were scanned and digitized using the NanoZoomer 2.0-HT Whole Slide Scanner (Hamamatsu Photonics; Japan). Tissue images were exported to a custom software program for cell counting (https://github.com/davepruitt/PRV-Cell-Counting; copy archived at https://github.com/elifesciences-publications/PRV-Cell-Counting). PRV-152 infected neurons expressed enhanced green fluorescent protein. PRV-152 neuron counts were made on every other forebrain and midbrain section (35 µm inter-slice interval) and every third spinal cord section (100 µm inter-slice interval). Experimenters performing analysis were blind to the group of each rat. Cortical neuron counts were restricted to layer 5 of sensorimotor cortex (*Bareyre et al., 2004*). We defined motor cortex using standard anatomical reference (*Paxinos and Watson, 2007*). Our ICMS mapping studies confirm that these regions contain the cortical forelimb sensorimotor circuitry. Red nucleus neuron counts were made using standard anatomical reference (*Paxinos and Watson, 2007*). Propriospinal neuron counts were made from spinal level C3 – C4 in Rexed lamina VI, VII, VIII and IX using standard anatomical reference similar to previous studies (*Watson et al., 2009*; *Gonzalez-Rothi et al., 2015*). Back-labeled putative spinal motor neurons were located in Rexed lamina IX and counted identical to previous studies (*Watson et al., 2009*; *Gonzalez-Rothi et al., 2015*). Sensorimotor cortex, red nucleus and cervical propriospinal neuron counts were normalized within rats to the number of putative spinal motor neurons in the lower cervical and upper thoracic spinal cord to control for any differences in injection efficacy. No differences in spinal neuron labeling were observed between CLV and Rehab alone (*Figures 2E* and *4E*). Sensorimotor cortex, red nucleus and putative spinal motor neuron counts were analyzed separately using unpaired t-tests. Data representing raw neuron counts in each ROI is available in *Figure 2—figure supplement 5*, *Figure 4—figure supplement 3*, and *Supplementary file 3*.

## Lesion Histology and Analysis

At the completion of experimental testing, rats were anesthetized with sodium pentobarbital (50 mg/kg, i.p.) and transcardially perfused with 4% paraformaldehyde in 0.1 M PBS (pH 7.5). The spinal cord was removed and spinal roots were kept for anatomical reference. Spinal tissue was then post-fixed overnight, cryoprotected in 30% sucrose for 48 hr, blocked and frozen at −80 C in Shandon M1 embedding matrix (Thermo Fisher Scientific; Waltham, MA). Spinal tissue was sliced at 50 µm using a cryostat, slide mounted and stained for Nissl (gray matter) and myelin (white matter) substance similar to previous studies (*Ganzer et al., 2016a*; *Ganzer et al., 2016b*). Photomicrographs were taken at 600 µm intervals to quantify gray and white matter lesion metrics using Image J. For unilateral SCI, the rostral and caudal extent of spinal gray and white matter damage was expressed as the percentage of spared gray and white matter of the right hemicord with respect to the left hemicord (*Figure 2—figure supplement 2* and *Figure 4—figure supplement 1*). For bilateral SCI rats, the rostral and caudal extent of spinal damage was expressed as the percentage of spared gray and white matter for each hemicord with respect to an unlesioned rostral and caudal tissue reference within animals (*Anderson et al., 2009*). Smallest and largest lesion areas were fitted to a schematic of spinal level C6 (*Figures 2E* and *3E*). To calculate damage to fiber tracts, two experimenters blind to group assignment evaluated the percentage of lesioned tissue to the dorsal corticospinal tract, dorsolateral corticospinal tract, ventral corticospinal tract, rubrospinal tract, propriospinal tract, and gray matter at the lesion epicenter. The dorsal, dorsolateral, and ventral corticospinal tracts were combined to calculate total CST damage based on the proportion of fibers in each tract (*Bareyre et al., 2005*). Data representing the damage estimates is available in *Supplementary file 4*.

## Data Availability

All source data supporting the findings of this study are available in the online version of the paper.

## Acknowledgements

We would like to thank Kim Rahebi, Abby Berry, Nikki Simmons, Preston D'Souza, Sanketh Kichena, Reshma Maliakkal, Ivan Rahman, Ami Shah, Kha Ai Hua, Luz Barron Horta, Zainab Haider, and Mian Bilal for technical assistance, and Michael Lane and Wayne Gluf for critical discussion, and Sven

Vanneste for assistance with statistics. PRV was a generous gift from Lynn Enquist. This work was supported by the WW Caruth, Jr. Foundation; NIH R01NS085167 and R01NS094384; Wings for Life Foundation; and by the Defense Advanced Research Projects Agency (DARPA) Biological Technologies Office (BTO) Electrical Prescriptions (ElectRx) program under the auspices of Dr. Eric Van Gieson through the Space and Naval Warfare Systems Center, Pacific Cooperative Agreement No. N66001-15-2-4057 and the DARPA BTO Targeted Neuroplasticity Training (TNT) program under the auspices of Dr. Tristan McClure-Begley through the Space and Naval Warfare Systems Center, Pacific Grant/Contract No. N66001-17-2-4011.

## Additional information

### Competing interests

Michael P Kilgard: has a financial interesting in MicroTransponder, Inc., which is developing CLV for stroke and tinnitus. The other authors declare that no competing interests exist.

### Funding

| Funder | Grant reference number | Author |
|---|---|---|
| Defense Advanced Research Projects Agency | N66001-15-2-4057 | Seth A Hays<br>Michael P Kilgard<br>Robert L Rennaker II |
| National Institutes of Health | R01NS085167 | April M Becker |
| Wings for Life | | Michael P Kilgard |
| W. W. Caruth Foundation | | Seth A Hays<br>Michael P Kilgard<br>Robert L Rennaker II |
| National Institutes of Health | R01NS094384 | Seth A Hays |
| Defense Advanced Research Projects Agency | N66001-17-2-4011 | Seth A Hays<br>Michael P Kilgard<br>Robert L Rennaker II |

The funders had no role in study design, data collection and interpretation, or the decision to submit the work for publication.

### Author contributions

Patrick D Ganzer, Conceptualization, Data curation, Formal analysis, Supervision, Validation, Investigation, Visualization, Methodology, Writing—original draft, Project administration; Michael J Darrow, Conceptualization, Data curation, Formal analysis, Validation, Investigation, Methodology, Writing—review and editing; Eric C Meyers, Formal analysis, Validation, Investigation, Visualization, Methodology; Bleyda R Solorzano, Andrea D Ruiz, Supervision, Investigation, Methodology; Nicole M Robertson, Justin T James, Han S Jeong, April M Becker, Investigation, Visualization, Methodology; Katherine S Adcock, Supervision, Investigation, Visualization, Methodology; Mark P Goldberg, Supervision, Funding acquisition; David T Pruitt, Software, Visualization; Seth A Hays, Conceptualization, Data curation, Formal analysis, Supervision, Funding acquisition, Validation, Writing—original draft, Project administration, Writing—review and editing; Michael P Kilgard, Conceptualization, Supervision, Funding acquisition, Validation, Writing—original draft, Project administration, Writing—review and editing; Robert L Rennaker II, Conceptualization, Supervision, Funding acquisition, Writing—original draft, Project administration, Writing—review and editing

### Author ORCIDs

Patrick D Ganzer http://orcid.org/0000-0003-4260-1624
April M Becker http://orcid.org/0000-0003-2706-1711
David T Pruitt http://orcid.org/0000-0001-8356-0887
Seth A Hays http://orcid.org/0000-0003-4225-241X
Robert L Rennaker II http://orcid.org/0000-0003-1260-1973

### Ethics

Animal experimentation: All procedures performed in the study were approved by the University of Texas at Dallas Institutional Animal Care and Use Committee. (Protocols: 14-10 and 99-06).

### Decision letter and Author response

Decision letter https://doi.org/10.7554/eLife.32058.031
Author response https://doi.org/10.7554/eLife.32058.032

## Additional files

### Supplementary files

• Supplementary file 1. Volitional Forelimb Strength Behavioral Data The values in the table below represent the average peak pull force for all rats included in the study at each week during therapy. Empty cells correspond to weeks in which data was not collected, based on the design of the experiment. All rat IDs are consistent for individual subjects throughout *Supplementary file 1–4*.
DOI: https://doi.org/10.7554/eLife.32058.025

• Supplementary file 2. Intracortical Microstimulation Data The values in the table below represent the area of motor cortex in mm (*Levy et al., 2016*) evoking movements in each of the categories. All rat IDs are consistent for individual subjects throughout *Supplementary file 1–4*.
DOI: https://doi.org/10.7554/eLife.32058.026

• Supplementary file 3. Pseudorabies Virus Labeling Data The values in the table below represent the total number of labelled eGFP+ neurons in each of the ROIs below. For rostrocaudal distribution of labeled neurons, see *Figure 2—figure supplement 3* and *Figure 4—figure supplement 5*. All rat IDs are consistent for individual subjects throughout *Supplementary file 1–4*.
DOI: https://doi.org/10.7554/eLife.32058.027

• Supplementary file 4. Lesion Metrics The values in the table below represent the estimated percent damage to gray matter and white matter tracts. All rat IDs are consistent for individual subjects throughout *Supplementary file 1–4*.
DOI: https://doi.org/10.7554/eLife.32058.028

• Transparent reporting form
DOI: https://doi.org/10.7554/eLife.32058.029

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
