## [Decision Letter]

Thank you for submitting your article "Closed-loop Neuroprosthesis Restores Network Connectivity and Motor Control after Spinal Cord Injury" for consideration by *eLife*. Your article has been favorably evaluated by Richard Ivry (Senior Editor) and three reviewers, one of whom, Heidi Johansen-Berg (Reviewer #1), is a member of our Board of Reviewing Editors. The following individual involved in review of your submission has agreed to reveal their identity: Reggie Edgerton (Reviewer #3).

The reviewers have discussed the reviews with one another and the Reviewing Editor has drafted this decision to help you prepare a revised submission.

Summary:

This manuscript reports the use of closed-loop vagus nerve stimulation to improve motor recovery after spinal cord injury. Overall, the results are potentially very interesting and impactful. However, the reviewers are concerned that the data are over interpreted in places and some key methodological details are lacking.

Essential revisions:

1) There were queries/concerns over the deliveries of reward. Some of the concerns may be cleared up as aspects of the methods are clarified. Specifically:

a) Trial structure is very confusing. Looking at Figure 1—figure supplement 1, it is not clear that reward was always given 50% of the time. For example, in G, reward appears to be given more frequently that 50%? Moving Figure 1—figure supplement 1 to Figure 1 with clear delineation of the trial structure, reward, etc. will help.

b) One reviewer commented that it appears that rats received an adaptive reward schedule, such that forces above a threshold resulted in a food reward. Therefore – top 20% pulls would be rewarded while bottom 20% pulls would not be rewarded. To what extent might this difference in reward contribute to results? The current interpretation is that the improved recovery for the top 20% group is due to selective stimulation of 'good' rather than 'bad' movements. However, it seems that the quality (force) of the movement is confounded by the reward. Perhaps if the bottom 20% movements had also been rewarded then equivalent recovery would have been seen in that group. Please clarify or else discuss this confound.

c) Another reviewer commented that it did not seem that the 120g criteria was used during rehab (i.e. as opposed to healthy training). It appears that during rehab the animals got a reward even if they remained at low levels of force. Is that correct? What was the distribution of forces that typically made the top/bottom 20% during rehab? Did they attempt a less adaptive approach that required higher force levels?

2) What period during rehab does Figure 1—figure supplement 1 represent? It is interesting that animals seem to start at low values and then rapidly 'ramp up' only with stimulation. In other words, example G appears to show an animal that does not ramp up rapidly without stimulation. Is there always an acute effect of VNS on the ability to ramp up to higher forces? Does the ability to generate higher forces during training explain the final sustained benefits?

3) The results reported in Figure 1 are used to support the notion that timing is critical. This is because the 0 and 2s delays are clearly better than the 25 second delay. However, in addition to the 25s condition having a longer mean delay – it is a much more variable delay (relative to the top 20% movements) as stimulation in this condition is not tied to the top 20% movement (but instead to the bottom 20% movements). Therefore there is confound between mean delay and variability of temporal coupling between movement and stimulation. I therefore suggest that interpretation of this result should be cautious. For example, the statement in the Discussion that 'We confirmed that activation of the vagus nerve must occur within seconds of a successful movement' is not adequately supported by the data. The data show that precisely timed stimulation within 2 seconds is effective, and that imprecisely timed stimulation at >20 seconds is not. That doesn't rule out the possibility that precisely timed stimulation at say 5 seconds may be effective.

4) Figure 1 are shown to illustrate similar number of trials and stimulated trials in the two stimulation groups. Although I can see that these distributions are overlapping I would not describe them as 'matched'. Trials and stimulations are reduced (albeit (presumably) not significantly so) in the bottom 20% group compared to the top 20% group. Demonstrating that these distributions don't' significantly differ from each other does not rule out the possibility that differences in trial numbers may have contributed to between group effects. To take into account differences in trial numbers would require, for example, including trial number as a confound in an ANCOVA.

5) It is great that the authors list the number of animals that were excluded (n=87); 87 is a rather large number. Please list more specifically why. How was device failure assessed?

---

## [Author Response]

Essential revisions:1) There were queries/concerns over the deliveries of reward. Some of the concerns may be cleared up as aspects of the methods are clarified. Specifically:a) Trial structure is very confusing. Looking at Figure 1—figure supplement 1, it is not clear that reward was always given 50% of the time. For example, in G, reward appears to be given more frequently that 50%? Moving Figure 1—figure supplement 1 to Figure 1 with clear delineation of the trial structure, reward, etc. will help.

We agree that the description of the reward and stimulation thresholds would benefit from additional clarification. To determine reward delivery, we utilized an adaptive reward threshold such that animals received a reward if the peak pull force on a trial exceeded the peak pull force of highest 50% (i.e., the median) of the preceding 10 trials *or* if it exceeded a fixed 120 g threshold. As a result, rewards could be delivered more often than 50% of the time, provided animals could generate pull forces greater than 120 g. We have used this capped adaptive threshold scheme in previous studies (1-3) This reward threshold was used in all groups at all times during the study.

We have provided additional text in the Materials and methods section (subsection “Vagus nerve stimulation parameters”) and legend for Figure 1—figure supplement 1 to clarify reward thresholds. Additionally, we have added stimulation thresholds to Figure 1 as suggested to further illustrate the adaptive thresholding.

b) One reviewer commented that it appears that rats received an adaptive reward schedule, such that forces above a threshold resulted in a food reward. Therefore – top 20% pulls would be rewarded while bottom 20% pulls would not be rewarded. To what extent might this difference in reward contribute to results? The current interpretation is that the improved recovery for the top 20% group is due to selective stimulation of 'good' rather than 'bad' movements. However, it seems that the quality (force) of the movement is confounded by the reward. Perhaps if the bottom 20% movements had also been rewarded then equivalent recovery would have been seen in that group. Please clarify or else discuss this confound.

The reviewer is correct in noting that an adaptive reward schedule was used. However, all groups received rewards based on the same capped adaptive reward threshold (described above). Thus, given that the reward schedule was identical across groups, reward delivery cannot account for the differences in recovery and plasticity observed across groups. We have now clarified the reward threshold in the Materials and methods section (subsection “Volitional Forelimb Force Generation Assessment”) and the legend for Figure 1—figure supplement 1.

c) Another reviewer commented that it did not seem that the 120g criteria was used during rehab (i.e. as opposed to healthy training). It appears that during rehab the animals got a reward even if they remained at low levels of force. Is that correct? What was the distribution of forces that typically made the top/bottom 20% during rehab? Did they attempt a less adaptive approach that required higher force levels?

The same adaptive reward threshold was in all groups at all times throughout the study, both during pre-training and during rehab after SCI. The capped adaptive reward threshold was set such that animals received a reward if the peak pull force on a particular trial exceeded the median peak pull force of the preceding 10 trials or exceeded a fixed 120 g threshold. Prior to injury, all animals easily exceeded the 120 g threshold on most trials, thus the reward threshold was typically stable at 120 g. The reviewer is correct that animals could receive a reward for lower levels of force. After injury, because of the substantial deficit in forelimb strength, animals very rarely exceeded the 120 g threshold. Thus, the vast majority of their rewards were delivered on trials that exceed the median of their previous ten trials, but fall below the 120 g threshold.

In general, following SCI, the bottom 20% of trials had pull forces of <16g, while pull force on the top 20% trials was >58g. The distributions of pull forces were similar for the Top 20% CLV and Bottom 20% CLV groups at the beginning of therapy. We now include a figure showing the distribution of pull forces for these groups (Figure 1—figure supplement 2).

We elected not to use a less adaptive approach, because our previous study demonstrated that fixed, static reward thresholds result in comparable performance, but a reduced number of trials, compared to an adaptive threshold (3). Therefore, in order to balance the number of trials performed across groups, we limited our study to use of the adaptive threshold.

2) What period during rehab does Figure 1—figure supplement 1 represent? It is interesting that animals seem to start at low values and then rapidly 'ramp up' only with stimulation. In other words, example G appears to show an animal that does not ramp up rapidly without stimulation. Is there always an acute effect of VNS on the ability to ramp up to higher forces? Does the ability to generate higher forces during training explain the final sustained benefits?

All examples are taken from the last week of rehabilitative therapy. The apparent ramp up is due to way in which thresholds are calculated. In all groups, the first ten trials of a session are rewarded regardless of pull force to encourage animals to perform the task when placed in the training apparatus. The reward threshold then adaptively scales (as described above) after the first ten trials, typically requiring greater pull forces to receive a reward. Because greater pull forces are required, performance typically does increase over the first 10-20 trials. This affect is independent of stimulation, as it is observed both prior to lesion and during rehabilitative training in animals that do not receive VNS (Rehab Alone). To provide a more comparable illustration with the other groups, we have replaced panel G with a different example.

Our previous studies demonstrate that there is no acute effect of VNS on performance (4, 5). In this study, no differences in performance are observed across groups after a single session of VNS, suggesting an absence of an acute effect (Peak Pull Force; One-way ANOVA, F[2,27] = 0.38, *P* = 0.69). Moreover, there was no trend toward increased pull forces within a session, nor differences in within-session pull force changes comparing groups that received VNS and groups that did not, at any time during therapy (Slope of peak pull forces over a session; One-way ANOVA, F[2,27] = 1.96, *P* = 0.16). Together, these findings confirm that VNS does not acutely change performance.

3) The results reported in Figure 1 are used to support the notion that timing is critical. This is because the 0 and 2s delays are clearly better than the 25 second delay. However, in addition to the 25s condition having a longer mean delay – it is a much more variable delay (relative to the top 20% movements) as stimulation in this condition is not tied to the top 20% movement (but instead to the bottom 20% movements). Therefore there is confound between mean delay and variability of temporal coupling between movement and stimulation. I therefore suggest that interpretation of this result should be cautious. For example, the statement in the Discussion that 'We confirmed that activation of the vagus nerve must occur within seconds of a successful movement' is not adequately supported by the data. The data show that precisely timed stimulation within 2 seconds is effective, and that imprecisely timed stimulation at >20 seconds is not. That doesn't rule out the possibility that precisely timed stimulation at say 5 seconds may be effective.

We agree with the reviewer that the temporal variability may also contribute to the reduced recovery observed with Bottom 20% CLV. Changes to the text have been made to temper the conclusions as suggested (Results, second paragraph and Discussion, first paragraph).

4) Figure 1 are shown to illustrate similar number of trials and stimulated trials in the two stimulation groups. Although I can see that these distributions are overlapping I would not describe them as 'matched'. Trials and stimulations are reduced (albeit (presumably) not significantly so) in the bottom 20% group compared to the top 20% group. Demonstrating that these distributions don't' significantly differ from each other does not rule out the possibility that differences in trial numbers may have contributed to between group effects. To take into account differences in trial numbers would require, for example, including trial number as a confound in an ANCOVA.

We agree with the reviewers that this qualification requires additional statistical support. We performed an ANCOVA on peak pull force and either trial number or stimulation number. After accounting for the number of trials, a significant group effect remains, consistent with the hypothesis that the number of trials cannot account for peak pull force (ANCOVA, effect of group: F[1,1] = 11.89, p = 0.0031). Similarly, a significant group effect on peak pull force remains after accounting for number of stimulations (ANCOVA, effect of group: F[1,1] = 9.57, p = 0.0066). We now include these statistical comparisons in the legend for Figure 1. Additionally, we have rephrased the statement to indicate that the groups performed a comparable, rather than matched, number of stimulations (Results, first paragraph).

5) It is great that the authors list the number of animals that were excluded (n=87); 87 is a rather large number. Please list more specifically why. How was device failure assessed?

In an effort to increase the rigor and reproducibility of our studies, we have taken care to diligently report any dropouts and their associated causes, thus we appreciate the reviewers’ recognition. The exclusion criteria match our previous studies (6-11). Given the large total number of animals in the study, the chronicity of the study, and the use of a neurostimulation device with associated failure modes, the 87 animals excluded is roughly equivalent to previous studies evaluating chronic recovery of motor function with VNS after neurological injury (6-10) We now provide more detail as to the specific reason for exclusion in the Materials and methods section (subsection “Experimental Design”). Additionally, we provide greater detail on the procedures used to assess device failure (subsection “Vagus nerve stimulation cuff implantation surgery”).

References

1) Ganzer, P. D. et al. Awake behaving electrophysiological correlates of forelimb hyperreflexia, weakness and disrupted muscular synchronization following cervical spinal cord injury in the rat. Behav. Brain Res. 307, 100–111 (2016).

2) Meyers, E. C. et al. Median and ulnar nerve injuries reduce volitional forelimb strength in rats. Muscle Nerve (2017).

3) Meyers, E. et al. The supination assessment task: An automated method for quantifying forelimb rotational function in rats. J. Neurosci. Methods 266, (2016).

4) Hulsey, D. R. et al. Reorganization of Motor Cortex by Vagus Nerve Stimulation Requires Cholinergic Innervation. Brain Stimul. 9, (2016).

5) Porter, B. A. et al. Repeatedly Pairing Vagus Nerve Stimulation with a Movement Reorganizes Primary Motor Cortex. Cereb. Cortex 22, 2365–2374 (2011).

6) Khodaparast, N. et al. Vagus nerve stimulation during rehabilitative training improves forelimb strength following ischemic stroke. Neurobiol. Dis. 60, (2013).

7) Khodaparast, N. et al. Vagus nerve stimulation delivered during motor rehabilitation improves recovery in a rat model of stroke. Neurorehabil. Neural Repair 28, 698–706 (2014).

8) Hays, S. A. et al. Vagus nerve stimulation during rehabilitative training enhances recovery of forelimb function after ischemic stroke in aged rats. Neurobiol. Aging 43, (2016).

9) Khodaparast, N. et al. Vagus Nerve Stimulation during Rehabilitative Training Improves Forelimb Recovery after Chronic Ischemic Stroke in Rats. Neurorehabil. Neural Repair 30, (2016).

10) Pruitt, D. T. et al. Vagus Nerve Stimulation Delivered with Motor Training Enhances Recovery of Function after Traumatic Brain Injury. J. Neurotrauma 33, 871–879 (2016).

11) Borland, M. S. et al. The Interval Between VNS-Tone Pairings Determines the Extent of Cortical Map Plasticity. Neuroscience (2017). doi:10.1016/j.neuroscience.2017.11.004